# Activin A Causes Muscle Atrophy through MEF2C-Dependent Impaired Myogenesis

**DOI:** 10.3390/cells11071119

**Published:** 2022-03-25

**Authors:** Audrey Loumaye, Pascale Lause, Xiaoling Zhong, Teresa A. Zimmers, Laure B. Bindels, Jean-Paul Thissen

**Affiliations:** 1Pole of Endocrinology, Diabetology and Nutrition, Institute of Experimental and Clinical Research, Université Catholique de Louvain, 1200 Brussels, Belgium; pascale.lause@uclouvain.be (P.L.); jeanpaul.thissen@uclouvain.be (J.-P.T.); 2Department of Endocrinology and Nutrition, Cliniques Universitaires Saint-Luc, 1200 Brussels, Belgium; 3Department of Surgery, Indiana University School of Medicine, Indiana University Simon and Bren Comprehensive Cancer Center, Richard L. Roudebush Veterans Administration Medical Center, Indianapolis, IN 46202, USA; xzhong@iu.edu (X.Z.); zimmerst@iu.edu (T.A.Z.); 4Research Service, Richard L. Roudebush Veterans Administration Medical Center, Indianapolis, IN 46202, USA; 5Metabolism and Nutrition Research Group, Louvain Drug Research Institute, Université Catholique de Louvain, 1200 Brussels, Belgium; laure.bindels@uclouvain.be

**Keywords:** activin A, skeletal muscle, muscle atrophy, myogenesis, MEF2C

## Abstract

Activin A (ActA) is considered to play a major role in cancer-induced cachexia (CC). Indeed, circulating ActA levels are elevated and predict survival in patients with CC. However, the mechanisms by which ActA mediates CC development and in particular skeletal muscle (SM) atrophy in humans are not yet fully understood. In this work, we aimed to investigate the effects of ActA on human SM and in mouse models of CC. We used a model of human muscle cells in culture to explore how ActA acts towards human SM. In this model, recombinant ActA induced myotube atrophy associated with the decline of MyHC-β/slow, the main myosin isoform in human muscle cells studied. Moreover, ActA inhibited the expression and activity of MEF2C, the transcription factor regulating *MYH7*, the gene which codes for MyHC-β/slow. This decrease in MEF2C was involved in the decline of MyHC-β/slow expression, since inhibition of MEF2C by a siRNA leads to the decrease in MyHC-β/slow expression. The relevance of this ActA/MEF2C pathway in vivo was supported by the parallel decline of *MEF2C* expression and SM mass, which are both blunted by ActA inhibition, in animal models of CC. In this work, we showed that ActA is a potent negative regulator of SM mass by inhibiting MyHC-β/slow synthesis through downregulation of MEF2C. This observation highlights a novel interaction between ActA signaling and MEF2C transcriptional activity which contributes to SM atrophy in CC models.

## 1. Introduction

Activin A (ActA), a member of the transforming growth factor-β (TGF-β) family, is expressed in a wide range of tissues and is involved in the regulation of multiple biological systems, such as embryogenesis, cell growth and differentiation, and immune and inflammation responses [1]. More recently, ActA has been implicated in cancer-induced cachexia (CC). Indeed, inhibitors of the activin-type IIB receptor (ActRIIB), which is shared with myostatin (Mstn), another member of TGF-β family, preserve skeletal muscle (SM) mass and prolong survival in murine models of CC [2,3,4,5,6,7]. In humans, elevated ActA is found in the bloodstream of patients with cancer disease and decreases after removal of cancerous tumor [8,9,10]. In addition, high circulating levels or tumor expression of ActA are associated with more aggressive and invasive tumors and the presence of metastasis [9,10,11,12,13,14,15,16,17]. Thereafter, we and others have shown that high circulating levels of ActA were associated with cachexia in cancer patients [18]. ActA levels correlated positively with the severity of cachexia and negatively with lean body mass [16,18,19,20,21,22]. In addition, ActA is predictive of survival in cancer patients [15,17,23]. Taken together, these results led us to hypothesize that ActA could influence survival of cancer patients by contributing to the development of cachexia and loss of SM mass. However, mechanisms through which ActA could mediate CC development and SM atrophy in particular are not yet fully understood.

Several experiments support the atrophying effect of ActA towards SM. Inhibin null mice, characterized by elevated circulating levels of ActA, develop a marked cachexia-like wasting syndrome associated with decreased survival [24]. This cachectic syndrome is characterized by an important loss of SM, heart, and fat mass. In the same way, the elevation of ActA levels locally or systemically, without underlying disease, leads to a dramatic decrease in SM mass and force, anorexia, and reduces life span [2,25,26,27,28,29]. Mechanistically, in vivo studies reveal that ActA exerts its effects on SM through the ActRIIB mainly by activating the pSmad2/3 pathway [26]. It results in an inhibition of Akt/mTOR pathway activation, leading to a slowdown of protein synthesis [26,27]. ActA also exerts a pro-fibrotic effect on SM [26]. In vitro studies have revealed that ActA impairs myogenesis and myotube differentiation [1,30]. Therefore, ActA alters SM mass and function by inducing SM atrophy, fibrosis, and impairing myogenesis.

To study the mechanisms of action of ActA on SM, the model used had to be considered. Indeed, although activins and Mstn are well conserved across species, their relative importance in the regulation of SM seems different across species. While a post-natal inhibition of Mstn induces a marked SM mass expansion in mice, small or no effects are observed on lean mass in healthy primates or humans [31,32,33]. In these species, only the inhibition of both at once, ActA and Mstn, provides a significant increase in lean mass. Interestingly, Latres et al. showed that while circulating levels of Mstn are higher (8-fold) in mice in comparison to monkeys and humans, circulating levels of ActA are higher (4-fold) in monkeys or humans in comparison to mice. Altogether, these data suggest that although Mstn is the main factor limiting SM growth in mice, ActA seems to have a predominant role in the regulation of SM in humans. All preclinical data are therefore not transposable to humans. For this reason, the effects of ActA on SM should be investigated in human muscle tissue or cells.

Given the elevation of circulating ActA levels during CC and its SM atrophying effect, well demonstrated in murine models, we aimed to investigate the effects of ActA on human SM. We used a model of human muscle cells in culture to explore how ActA acts towards human skeletal muscle. A better knowledge of mechanisms involved in ActA-induced SM atrophy in humans could provide a new target for therapeutic intervention, potentially useful in the care of CC.

## 2. Experimental Procedure

### 2.1. Cell Culture, Treatment, and Transfection

Adult human skeletal muscle-derived cells (SkMDC) obtained from a 41-year-old donor (Cook Myosite, Pittsburgh, PA, USA) were cultured in growth medium consisting of DMEM with GLUTAMAX™, 20% FBS, 0.5% Ultroser G (Pall, Cergy, France), 1% antibiotic-antimycotic, and 1% nonessential amino acids at 37 °C in a 5% CO_2_ incubator. SkMDC mean population doubling was determined at each passage and cells were used before the thirteenth division. After 4 days of proliferation, when the seeding density reached 70–80%, growth medium was replaced by the fusion medium consisting of DMEM with GLUTAMAX™, 2% horse serum, 1% FBS, 1% antibiotic-antimycotic, and 1% nonessential amino acids. All the experiments were performed in triplicates in six-well plates and independently replicated three or five times as indicated (n = 3 or 5).

After 4–5 days of differentiation, SkMDC were treated with human recombinant ActA (100 ng/mL or as indicated) (R&D Systems, Abingdon, UK) or by vehicle (HCl 4 mM) (Ctrl) for 48 h. The fusion medium with ActA or the vehicle was renewed after 24 h.

For small interfering RNA (siRNA) experiments, SkMDC were transfected after 4 days of differentiation with either a specific On-Targetplus SiRNA SMART-pool against human MEF2C (25 nM) (SiMEF2C) or a negative control (On-Targetplus non-targeting pool SiRNAs) (25 nM) (SiRNA [−]) (all from Dharmacon, Thermo Fisher Scientific, Waltham, MA, USA). The fusion medium was renewed after 24 h.

For miR mimic experiments, cells were transfected after 5 days of differentiation with a synthetic mimicking mature endogenous miR-1 (miR-1 mimic, 10–100 nM, from Qiagen) or negative control (On-Target plus non-targeting pool) (25 nM). Transfection was performed by using Lipofectamine^®^ RNAiMAX Reagent (Life Technologies, Carlsbad, CA, USA). Cells were collected 48 h post-transfection. Cells were rinsed twice in phosphate buffered saline (PBS) before RNA and protein extraction.

### 2.2. Cell Viability

Cell viability was determined using the CellTiter-Glo^®^ luminescent cell viability kit (Promega Corporation, Madison, WI, USA) in accordance with the manufacturer’s instructions. This method is based on the measurement of ATP production by the cells, proportional to the number of viable cells, detected by luciferin–luciferase reaction.

### 2.3. Myotube Morphological Analysis

Myotubes were labeled with mouse monoclonal anti-myosin-heavy chain (MyHC) antibody (MF20, 1:20 dilution) and resolved with secondary antibodies conjugated to Alexa-Fluor 488 (1:400 dilution) (Invitrogen, Cergy-Pontoise, France). Images were captured with a high-resolution cooled digital XC30 camera coupled to a BX-50 microscope (Olympus, Rungis, France) at a resolution of 0.64 μm/pixel. The myotube diameter was measured on 100 myotubes in each condition (from 3 independent experiments). Myotubes were defined by the presence of minimum 5 nuclei. For each myotube, five random measurements were performed along the length of the myotube using the image processing software ImageJ 1.47v, and the average of these five measurements was considered as one single value.

### 2.4. Direct miRNA or mRNA Quantification by RT-qPCR

Total RNA and miRNA were extracted from cultured myotubes or frozen muscle samples using TriPure Isolation Reagent (Roche Diagnostics, Basel, Switzerland), as described by the manufacturer. Reverse transcription and real-time quantitative PCR were performed as previously described [34]. Relative mRNA levels were calculated using the comparative threshold cycle (Ct) method and normalized by the expression of the housekeeping gene *GAPDH* for in vitro data, and *Cyclophiline* or *Tbp* for in vivo data. The primer sequences used for amplification during real-time qPCR are listed in Table 1. For miR quantification, 1 μg of total RNA was reverse transcribed by using the miScript II RT PCR kit (Qiagen, Hilden, Germany), and 10 ng of total RNA equivalent was amplified with miScript Syb Green PCR kit (Qiagen, Hilden, Germany) using commercial miRNA-specific forward primers (Qiagen, Hilden, Germany) and a reverse universal primer (provided in the miScript II RT PCR kit).

### 2.5. Western Blotting

Myotube proteins were homogenized in ice-cold pH 7.0 buffer containing 50 mM of Tris-HCl, 150 mM of NaCl, 5 mM of EDTA, 2% NP40, 0.1% SDS, 1 mM of phenylmethylsulfonyl fluoride, 10 μg/mL of leupeptin, 10 μg/mL of aprotinin, and 1 mM of sodium orthovanadate. Homogenates were centrifuged at 16,000× *g* for 20 min at 4 °C, and supernatants were immediately stored at 80 °C. Fifteen micrograms of muscle cells proteins were resolved by SDS polyacrylamide gel 10% electrophoresis and transferred to PVDF membranes. Membranes were probed with the following primary antibodies: pSmad2, Smad2/3, and MEF2C (all from Cell signaling Technology, Leiden, The Netherlands) and MyHC-β/slow (Sigma-Aldrich). Signals were revealed by Enhanced Chemiluminescence^®^ Western Blotting Detection Plus (GE Healthcare, Machelen, Belgium), then quantified and normalized to total protein loading assessed by staining membranes using Coomassie blue.

### 2.6. Mouse Models of Cancer Cachexia

#### 2.6.1. C26 and Baf3 Models

Mice from Charles River Laboratories (Chatillon-sur-Chalaronne, France) were housed at two mice per cage with a 12-h light/dark cycle and fed an irradiated normal chow diet (AO4-10, 2.9 kcal/g, Safe, Augy, France). CD2F1 male mice (8-week-old) were injected subcutaneously (intrascapular) with C26 cells (1 × 10^6^ cells in 0.1 mL of saline, C26 group; n = 8) or a saline solution (Ctrl group; n = 8), as previously described [35]. Balb/c female mice (6-week-old) were injected into the tail vein with BaF3 cells (1 × 10^6^ cells in 0.1 mL of saline, BaF3 group; n = 8) or saline solution (Ctrl group; n = 8) after anesthesia, as previously described [35]. At the end of the experiment (10 days for the C26 model and 14 days for the BaF model), tissue samples were harvested following anesthesia (ketamine and xylazine, or isoflurane gas, Abbot, Wavre, Belgium). Tissues were weighed and frozen in liquid nitrogen. All the samples were stored at −80 °C. The experiments were approved by and performed in accordance with the guidelines of the local ethics committee from the Université catholique de Louvain. Housing conditions were as specified by the Belgian Law of 29 May 2013, regarding the protection of laboratory animals (Agreement No. LA1230314).

#### 2.6.2. KPC Model

Under a protocol approved by the Institutional Animal Care and Use Committee of Indiana University School of Medicine, KPC and age-matched genotype control mice were housed up to five per cage in a pathogen-free facility on a 12-h light cycle, with *ad libitum* access to autoclaved food and sterile water. KPC (Kras^G12D^; Trp53^R172H^; Pdx1-Cre) mice are a genetically engineered mouse model that mimics human PDAC (pancreatic ductal adenocarcinoma) [14]. For activin inhibition, early-stage male KPC mice with tumors measuring 5–7 mm and body condition score from 3–6 were treated with either ACVR2B/Fc (15 mg/kg body weight i.p.) or the same volume of phosphate buffered saline (vehicle control), administered every 5 days until euthanasia. Mice were euthanized when tumors reached >10 mm and the body condition score ≥ 8. The body condition score was determined by summing the points from each domain as follows: body posture (0—normal, 1—mildly hunched, 2—moderately hunched, 3—very hunched); activity level (0—normal, 1—slightly reduced, 2—moving slowly, 3—moving reluctantly or not at all); and eye appearance (0—eye closed <25%, 1—eye closed 25–50%, 2—eye closed 51–75%, 3—eye closed 76–100%). Endpoints in KPC mice were compared with no-tumor, genotype controls of age-matched siblings, co-housed with KPC mice. Tissues were collected and weighed, then snap frozen in liquid nitrogen for protein or RNA extraction. Frozen plasma and tissue were stored at −80 °C. Activin A levels were determined by activin A Quantikine ELISA (R&D Systems, Abingdon, UK). This was a sub-study of a larger study currently in press (Zhong, X., Narasimhan, A., Silverman, L.M., Young, A.R., Shahda, S., Liu, S., Wan, J., Liu, Y., Koniaris, L.G., Zimmers, T.A. Sex specificity of pancreatic cancer cachexia phenotypes, mechanisms, and treatment in mice and humans—Role of Activin A. JCSM, in press).

### 2.7. Statistical Analysis

The results are presented as means ± SEM for the indicated number of cell experiments or mice. Statistical analyses were performed using an unpaired *t*-test to compare two conditions or one-way ANOVA to compare three or more conditions. Statistical analyses were performed using GraphPad Prism 7 Software, San Diego, CA, USA). Significance was set at *p* < 0.05. *, *p* < 0.05, **, *p* < 0.01, and ***, *p* < 0.001.

## 3. Results

### 3.1. Activin A Causes Atrophy of Human Skeletal Muscle Cells

To explore the effect of ActA on human SM, primary well-differentiated human myotubes were exposed to recombinant ActA (100 ng/mL) for 48 h. In these conditions, ActA caused a significant decrease in myotube diameter (−21%, *p* = 0.010) (Figure 1A). The effect of ActA on the size of myotubes was dose-dependent (Figure 1A). The chosen dose of ActA (100 ng/mL) was the optimal dose to obtain a significant myotube atrophy without altering cell viability. The myotube atrophy caused by ActA was associated with a marked increase in Smad2 phosphorylation already observed after one hour of treatment (Figure 1B). ActA therefore caused atrophy of human-differentiated myotubes, mediated by an activation of the Smad2/3 pathway, as already shown for Mstn [30].

### 3.2. Activin A-Induced Myotube Atrophy Is Characterized by a Decrease in Myosin-Heavy Chain-β/Slow Content

Since contractile proteins myosin and actin are the two most abundant structural proteins in SM cells, we investigated the consequences of ActA treatment on myosin-heavy chains (MyHC) and their gene expression. Interestingly, ActA led to a decrease in MyHC-β/slow content, the most abundant myosin isoform in slow muscle fibers (−44%, *p* = 0.013) (Figure 2A). This decline in MyHC content was caused by a decreased expression of the *MYH7* gene (−45%, *p* = 0.0001), encoding the MyHC-β/slow, whereas no significant changes were observed in the expression of *MYH1* or *MYH2* genes, encoding, respectively, MyHC-2X and MyHC-2A, the most abundant myosins in fast fibers (Figure 2B) [36,37]. In contrast to mice, human SM are composed mainly (>90%) of slow and fast 2A fibers [38]. In our myotubes, obtained from rectus abdominis muscle, *MYH7* expression was predominant and was 7 and 70 times more expressed than *MYH1* or *MYH2*, respectively (Figure 2C). Accordingly, ActA therefore induced a decrease in gene expression and protein content of MyHC-β/slow, the most abundant myosin isoform in human SM cells studied.

### 3.3. Activin A-Induced Myotube Atrophy Is Associated with a Downregulation of MEF2C Expression and Activity

Since MEF2C is the main transcription factor regulating the expression of *MYH7*, the gene encoding for MyHC-β/slow protein, we characterized the effect of ActA on MEF2C. ActA caused a marked decrease in MEF2C protein content (−85%, *p* = 0.0497) and its gene expression (−48%, *p* = 0.020) (Figure 3A,B). To investigate if a downregulation of MEF2C expression led to a decrease in its activity, we quantified mRNA levels of its main targets. As we observed, ActA caused a decrease in mRNA levels of *MB* (−51%, *p* = 0.041), *MYOM-1* (−42%, *p* = 0.0006) and *PPARGC1A* (−55%, *p* = 0.041), encoding, respectively, for myoglobin, myomesin-1, and peroxisome proliferator-activated receptor gamma coactivator 1-alpha (PGC1-α) (Figure 3C). ActA is known to decrease some SM-specific microRNAs or myomiRs (miRs) involved in SM development, regeneration, and phenotype. We observed a significant decrease in miR-1 (−82%, *p* = 0.015), which is also a target of MEF2C (Figure 3D).

To identify the factors responsible for the downregulation of MEF2C expression by ActA, we quantified mRNA level of upstream myogenic regulating factors (MRFs), such as MyoD, myogenin, and Myf5 [38,39,40,41]. ActA induced a significant downregulation of *MyoD* (−50%, *p* < 0.028) and *MyoG* (−51%, *p* = 0.001) without affecting *Myf5* expression (Figure 3E). Since MEF2C activity is also regulated by myogenic regulatory factor (MRF)4 and histone deacetylase (HDAC)4, we investigated their expression in response to ActA. ActA caused no significant change in *MRF4* or *HDAC4* expression (Figure 3E).

Therefore, the decrease in MyHC-β/slow synthesis caused by ActA is associated with the downregulation of expression and activity of MEF2C, the main transcriptional factor of *MYH7*.

### 3.4. MEF2C Is Required to Maintain MyHC-β/Slow Gene Expression and Protein Content in Differentiated Myotubes

The parallel downregulation of *MEF2C* and *MYH7* gene expression in response to increasing concentrations of ActA (Figure 4A,B) supports the role of decreased *MEF2C* in the decline of *MYH7* in our model. 

To highlight the obligatory role of MEF2C in the *MYH7* transcription, we examined the impact of the inhibition of MEF2C by a specific siRNA on MyHC-β slow content. Seventy-two hours after transfection, MEF2C silencing was effective as demonstrated by the severe decrease in MEF2C mRNA (−94%, *p* = 0.0002) and protein level (−100%, *p* = 0.009) (Figure 4C–E). The abolition of MEF2C led to the decrease in slow MyHC-β/slow content (−48%, *p* = 0.009) (Figure 4C,F,G), pointing out the crucial role of MEF2C as a major transcriptional regulator of *MYH7*, and therefore involved in the maintenance of MyHC-β/slow content.

We then investigated whether forced expression of MEF2C could increase MyHC-β/slow expression in differentiated myotubes. To force the expression but also the activity of MEF2C, we overexpressed miR-1 in myotubes. Indeed, miR-1 is a muscle-specific miR, particularly expressed during differentiation of myoblasts into myotubes. MiR-1 is involved in myogenesis by positively regulating the expression of myogenic factors, such as MyoD, myogenin, and MEF2C. On the one hand, miR-1 targets HDAC4, the main repressor of MEF2C activity, and on the other hand, miR-1 is also a transcriptional target of MEF2C, leading to a positive regulation loop of MEF2C activity [42]. Given these observations, we investigated whether overexpression of miR-1 could increase MEF2C and MyHC-β/slow content. The transfection of miR-1 mimic in differentiated myotubes was effective, as illustrated by the increased expression of miR-1 (Figure 4H). More importantly, overexpression of miR-1 induced a dose-dependent increase in MEF2C and MyHC-β/slow content (Figure 4I). MiR-1 was therefore sufficient to induce MEF2C expression and activity together with an increase in MyHC synthesis, strengthening the link between MEF2C and MyHC-β/slow content.

### 3.5. The Activin A-Induced Myotube Atrophy Is Not Associated with Increased Classical E3 Ubiquitin Ligases

The SM upregulation of different E3 ubiquitin ligases, also called atrogenes, such as TRIM63, Atrogin1, or MUSA1, has been highlighted in several models of SM atrophy, in particular in CC [43]. The ActA-induced reduction of MyHC-β/slow content could be due to a rise in degradation by proteolysis. However, whether an elevation of proteolysis or expression levels of atrogenes is observed in ActA-induced SM atrophy remains controversial. Inconsistencies in results observed in studies seem to be dependent on the model used. To clarify ActA effects on regulation of atrogenes expression in human SM, we quantified the mRNA level of the most characterized ones, namely TRIM63, Atrogin1, and MUSA1 in myotubes, 48 h after treatment with recombinant ActA (100 ng/mL). ActA induced a decrease in *TRIM63* expression (−34%, *p* < 0.05), an increase in Atrogin1 expression (+55%, *p* < 0.05), and no changes in MUSA1 expression (Figure 5). Consistent with in vivo models, ActA does not upregulate atrogenes targeting the MyHC, such as TRIM63. However, ActA upregulates Atrogin1, which targets factors regulating protein synthesis. Taking together, these observations do not support a stimulation of MyHC proteolysis by ActA.

### 3.6. Cancer Cachexia Is Associated with Downregulation of Muscle MEF2C Expression and Activity Which Is Blunted by Inhibition of Activin A

Given the potential role of ActA in CC [2,18,26,27] and its inhibitory effect on MEF2C expression, we investigated the muscle expression of MEF2C in animal models of CC. The C26 model is a well-described model of CC caused by subcutaneous implantation of colon adenocarcinoma cells and is associated with elevated levels of circulating ActA [27], partially due to high tumoral expression of *INHBA* (data not shown). As expected, C26 mice exhibited a marked body weight loss and muscle atrophy, as demonstrated by the decrease in muscle weight (−18%; *p* < 0.0001) (Figure 6A). In atrophied muscle, the *MEF2C* mRNA level was profoundly decreased (−58%; *p* < 0.0001) together with a decreasing trend in the *MYH7* mRNA level (−36%; *p* = 0.051) (Figure 6B–E). To extend the observations to other CC animal models, we quantified the muscle expression of these genes in an acute leukemia model (BaF3 model). Consistent with our results obtained in C26 mice, we observed a dramatic decrease in *MEF2C* (−94%; *p* < 0.0001) and *MYH7* (−53%; *p* < 0.0001) mRNA levels in atrophied muscles of BaF3 mice (Figure 6G–J). In both CC models, myogenesis was impaired, as supported by the changes in *MyoD* and *MyoG* expression (data not shown), while *MYH1* and *MYH2* expression was decreased (Figure 6C,D,H,I) as observed in previous works [44,45,46].

We then investigated whether the anti-atrophic effect exerted by ActA inhibition in CC models is associated with restoration of MEF2C expression [2,14]. To answer this question, we systemically administrated an inhibitor of ActA activity (ACVR2B/Fc) or phosphate-buffered saline (Ctrl) to male genetically engineered KPC mice bearing autochthonous tumors. KPC mice exhibited a huge elevation of circulating ActA levels, which were maintained after the administration of ACVR2B/Fc (Figure 6K). As expected, KPC mice developed an SM atrophy, as demonstrated by the decrease in muscle weight (−32%; *p* < 0.0001), in association with a downregulation of *MEF2C* muscle expression (−59%; *p* = 0.0015) (Figure 6L,M). Interestingly, the inhibition of ActA activity led to preserving muscle mass, together with *MEF2C* muscle expression in KPC mice, without effects on tumor growth (Figure 6L–N).

Taken together, our in vivo data show that *MEF2C* expression and its target, *MYH7*, are downregulated in animal models of CC. In addition, the tumor-induced elevation of circulating ActA levels seems to be involved in the decrease in *MEF2C* muscle expression, since the inhibition of ActA activity allowed muscle mass and MEF2C expression to be preserved, supporting the relevance of our in vitro observations.

## 4. Discussion

In the present work, we characterized the effects of ActA on human-differentiated myotubes. We showed that ActA induces human muscle cell atrophy, which is associated with the decline in MyHC-β/slow content, the main myosin isoform in the human muscle cells studied. The decline of MyHC-β/slow content results from inhibition of its synthesis caused by a downregulation of *MYH7* mRNA. MEF2C and miR-1, the two main regulators of *MYH7* expression, are inhibited by ActA, which suggests their role in the decreased *MYH7* mRNA. Indeed, MEF2C is mandatory to maintain *MYH7* and MyHC-β/slow expression in differentiated myotubes. The relevance of this pathway in vivo is supported by the parallel decline of *MEF2C* expression and muscle mass, which are both inhibited by an ActA antagonist in animal models of CC. Our observation therefore highlights a new signaling pathway by which ActA causes muscle atrophy, in particular during CC.

### 4.1. Activin A Causes Human Muscle Cell Atrophy by Altering Myogenesis and MyHC-β/Slow Synthesis

We showed that ActA induces human SM cell atrophy together with the decline in MyHC-β/slow content and the downregulation of several MRFs, such as MyoD and myogenin, as already shown by others in muscle cells exposed to Mstn [30,47,48]. Since these MRFs are mandatory for the differentiation of myoblasts to myotubes, our results suggest that myotube atrophy caused by ActA results from blunting of the myogenesis [49]. Myogenesis is the process that leads to SM formation during embryogenesis. In adults, satellite cells remain quiescent and are activated to proliferate and differentiate for SM regeneration in case of injury [37]. More recently, it has been suggested that alterations in the regenerative process might contribute to SM wasting during catabolic situations, such as CC [45,49]. While ActA negatively regulates SM mass to avoid excessive growth during development, a systemic or local increase in ActA could impair the regeneration process necessary to maintain SM mass during catabolic situations, leading to SM atrophy [1]. Impaired myogenesis in response to ActA has already been reported by previous studies. In vitro, ActA alters myogenesis by inhibiting the differentiation of myoblasts to myotubes in murine and human muscle cells [30,47]. In addition, ActA may mediate the inhibition on myogenesis caused by pro-inflammatory cytokines such IL-1α and TNF-α [50]. However, our results indicate that ActA may inhibit the myogenesis program, even in well-differentiated myotubes. As demonstrated for Mstn in human myotubes, it is likely that this inhibition results from activation of the Smad pathway (30). Therefore, the SM cell atrophy that we observed in human myotubes exposed to ActA may result from inhibition of myogenesis. Although we found mainly a decrease in MyHC-β/slow content in response to ActA, others reported a decrease in both fast and slow MyHCs in human-differentiated myotubes [51]. The discrepancy between these data and ours could result from differences in the nature of the muscle used for the primary culture and from the interindividual variability which influences the phenotype, namely the proportion of fast or slow fibers in the original muscle. Our data do not exclude the possibility that ActA downregulates the expression of other MyHCs than the MyHC-β/slow, depending on the starting phenotype. Therefore, additional studies on a large number of subjects would be necessary to clearly characterize the effect of ActA on different types of fibers.

### 4.2. Activin A Inhibits MyHC-β/Slow Synthesis by Downregulating MEF2C

By seeking to identify the mechanism responsible for the inhibition of MyHC-β/slow synthesis in response to ActA, we were able to demonstrate a decrease in *MEF2C* expression, a transcription factor cardinal for the *MYH7* expression, the gene encoding MyHC-β/slow. MEF2 is a transcription factor of the MADS box family, which is involved in the development of several human cells, including muscle, neural, chondroid, immune, and endothelial cells [42]. The MEF2 family includes four factors: MEF2A, B, C, and D. MEF2A and C, the most expressed in SM, play a major role in myogenesis during development and regeneration and in myofiber phenotype determination by promoting a slow gene program [42]. In particular, the transcriptional activity of MEF2A and C is essential for regeneration [52]. As MEF2 factors and MRFs, in particular MyoD and myogenin, regulate each other’s expression in positive feedback loops, ActA could decrease *MEF2C* expression by downregulating these upstream MRFs, as we observed [42]. Furthermore, it is worth noting that MEF2 acts synergistically with MRFs to regulate muscle-specific gene expression [41].

Not only expression but also activity of MEF2C is decreased in human myotubes exposed to ActA. This conclusion is supported by the parallel decrease in MEF2C and its main targets, *MYH7*, but also *MB*, *MYOM-1*, and *PPARGC1A*, encoding, respectively, for MyHC-β/slow, myoglobin, myomesin-1, and PGC1-α, and myomiR, such as miR-1 (Figure 7). These changes caused by ActA might impair structural but also metabolic capacities of differentiated myotubes. Indeed, myomesin-1 is a protein localized in M-lines and involved in structural organization of the sarcomere, myoglobin plays a major role in oxygen storage and diffusion in myotubes, whereas PGC1-α is involved in mitochondrial biogenesis and fiber-type determination [38,53,54].

The crucial role of MEF2C for the synthesis of MyHC-β/slow was clearly demonstrated by two observations that we made. First, the transfection of myotubes with miR-1 mimic led to a clear dose-dependent increase in MEF2C and MyHC-β/slow content. Second, transfection of siRNA targeting MEF2C in myotubes led to a decrease in MEF2C and MyHC-β/slow content. MEF2C appears therefore as an essential factor regulating the transcription of slow-fiber-specific genes, in particular *MYH7* and its product MyHC-β/slow.

Although *MEF2C* knock down decreased *MYH7* expression and MyHC-β/slow content, we could not demonstrate that overexpression of MEF2C restores MyHC-β/slow content in myotubes exposed to ActA (data not shown). These results suggest that ActA impairs MEF2C activity, whatever its expression. Different hypotheses could be proposed to explain how ActA inhibits MEF2C transcriptional activity. Firstly, MEF2C acts in combination with several co-factors which are potentially also downregulated by ActA and which are required in sufficiency to insure the MEF2C transcriptional activity. Secondly, independently of its expression, MEF2C activity may also be inhibited directly by the Smad pathway, which is stimulated by ActA [55,56]. Indeed, Smad3 can suppress the function of MEF2, by disturbing the synergy between MyoD and MEF2 and by preventing its association with its coactivators, such as GRIP-1 [56]. Alternatively, Smad3 may favor the MEF2 export from the nucleus to the cytoplasm, making MEF2 unavailable to transcriptional complex and preventing transcription of downstream targets, a mechanism used by TGF-β1 which also inhibits myogenesis through Smad activation [55]. Taken together, these data support that ActA downregulates not only MEF2C expression but could also repress MEF2C activity via Smad2/3 signaling. Further experiments will be necessary to delineate whether forced MEF2C, specifically in nuclei of myotubes, could protect SM cells against ActA atrophic effects. Nevertheless, in tumor-bearing mice, the preservation of SM mass induced by the inhibition of ActA is associated with the preservation of muscle *MEF2C* expression. This observation strengthens the link between the elevation of ActA and MEF2C downregulation in cancer-induced SM atrophy.

### 4.3. Activin A Targets miR-1 Expression, a Positive Regulator of MEF2C Expression and Activity

The regulation of MEF2 activity is complex and involves not only upstream MRFs, but also some miRs. By seeking the contribution of some miRs for the inhibition of MyHC-β/slow synthesis by ActA, we were able to demonstrate a decrease in miR-1. MiRs are small, noncoding RNA, which post-transcriptionally regulate gene and consequently protein expression. MyomiR are miRs specifically expressed in SM and involved in the regulation of SM development, regeneration, and phenotype [37,38]. Among them, miR-1 is a conserved muscle-specific miR, particularly expressed during the differentiation of myoblasts into myotubes [57]. Our attention was focused on miR-1 for at least two reasons. First, miR-1 is involved in myogenesis by positively regulating the expression of myogenic factors, such as MyoD and MEF2 [57]. Among myogenic factors, MEF2C has a specific interaction with miR-1, since miR-1 is a transcriptional target of MEF2C but also targets HDAC4, the main repressor of MEF2C activity [42,57]. Consistent with these data, we showed that overexpression of miR-1 in human myotubes increases MEF2C expression and activity, as supported by the increase in MyHC-β/slow content. Second, miR-1 has previously been shown to be regulated by ActA/Mstn signaling, which suggests its role in the regulation of muscle mass. Whereas increased miR-1 is associated with muscle hypertrophy caused in vivo by Mstn deletion (unpublished) and in vitro by ActA/Mstn inhibition [58], decreased miR-1 is observed in situations of muscle atrophy induced by Mstn in C_2_C_12_ myotubes [58]. The decrease in miR-1 that we observed in response to ActA is not so unexpected, since miR-1 is positively regulated by myogenic factors (MyoD and MEF2C) and the mTOR pathway, both known to be repressed by ActA [57,59].

### 4.4. Activin A Does Not Upregulate Classical E3 Ubiquitin Ligases Targeting Myosin-Heavy Chain

To investigate the possibility for ActA to stimulate proteolysis of MyHC-β/slow, we quantified the expression of the main atrogenes in human myotubes. In our conditions, ActA negatively regulates TRIM63, but stimulates Atrogin1, and has no effect on MUSA1. The opposite regulation of these atrogenes by ActA is not so surprising since their targets are different. Whereas TRIM63 targets structural proteins such as MyHC, actin, troponin I, and myosin light chain, Atrogin1 targets factors regulating cell growth and protein synthesis, such as MyoD and eIF3f [43]. Previous observations support our results. Indeed, systemic elevation of ActA increases Atrogin1 expression without any changes in TRIM63 expression in vivo [26]. Similarly, human myotubes exposed to Mstn exhibit a dose-dependent downregulation of TRIM63 and Atrogin1 [30]. All together, these data suggest that accelerated structural protein degradation is not the main mechanism responsible for SM atrophy caused by ActA/Mstn signaling.

### 4.5. The Downregulation of MEF2C Expression and Activity in Animal Models of Cancer Cachexia Is Reversed by an Activin A Antagonist 

The inhibition of MEF2C transcriptional activity by ActA that we identified in myotubes is particularly relevant for CC, a situation characterized by high levels of ActA in mice and in humans [14,18,20,27]. Indeed, we observed a marked downregulation of *MEF2C* and *MYH7* expression in atrophied muscle of animal models of CC. In addition, the preservation of SM mass, induced by inhibition of ActA, is associated with the preservation of muscle MEF2C expression in tumor-bearing mice. These observations strengthened the link between the elevation of ActA and MEF2C downregulation in cancer-induced SM atrophy. The contribution of low MEF2C expression to the SM atrophy observed in CC is supported by several observations. The loss and gain of function studies reveal indeed that MEF2C play a role in the control of SM mass and fiber-type determination [41]. Constitutional muscle-specific deletion of MEF2C leads to major hints in SM development and early lethality. However, if the deletion occurs in adulthood, mice exhibit a low proportion of slow fibers [38,39]. By contrast, overexpression of MEF2C increases the expression of PGC-1α and the proportion of slow fibers [38]. Recently, a downregulation of MEF2C muscle expression was observed in other conditions associated with SM atrophy, such as fasting and unloading [60,61,62], suggesting that downregulation of MEF2C could be a generalized mechanism involved in SM atrophy. Even more interesting, overexpression of MEF2C or an increase in its activity causes SM hypertrophy and prevents SM atrophy in the case of CC and denervation [61,63]. Together with our data, these observations indicate that the role of MEF2C goes well beyond the myogenesis and regeneration, and may touch more broadly on the SM mass control. Taken together, these data provide new potential therapeutic targets to treat SM atrophy or impaired regeneration occurring during cancer-induced cachexia.

Although previous studies in animals highlighted a preferential atrophy of fast fibers in cancer cachexia, we reported in our two preclinical models a decrease in the *MYH7* which characterizes slow fibers [2,64]. However, our focus on MyHC-β/slow must not omit the fact that fast MyHCs are also decreased in these models, in agreement with a recent review [5] which concluded that cancer cachexia is associated with a decrease in the size of slow and fast fibers in both mice and humans. Although our work investigated the potential role of MEF2C in slow fiber atrophy during cancer cachexia, the atrophy of fast fibers and the role of factors other than ActA to explain it is more than likely.

## 5. Conclusions

Although ActA actions towards SM have widely been assessed in *in vivo* studies, and particularly in preclinical models of CC, few data exist regarding the direct effects and mechanisms of actions of ActA on muscle cells. In addition, given the differences across species regarding the respective role of Mstn and ActA toward skeletal muscle, we used a model of primary culture of human cells to investigate the effects of ActA in humans. In this work, we showed that ActA is a potent negative regulator of SM mass by causing inhibition of MyHC-β/slow synthesis. This resulted from the downregulation of MRFs and MEF2C directly involved in the transcription of *MYH7*, the gene coding for MyHC-β/slow, the main myosin isoform in human muscle studied. We identified therefore a novel interaction between ActA/pSmad2/3 signaling and MEF2C transcriptional activity that could mediate SM cell atrophy in response to ActA.

## Figures and Tables

**Figure 1 cells-11-01119-f001:**
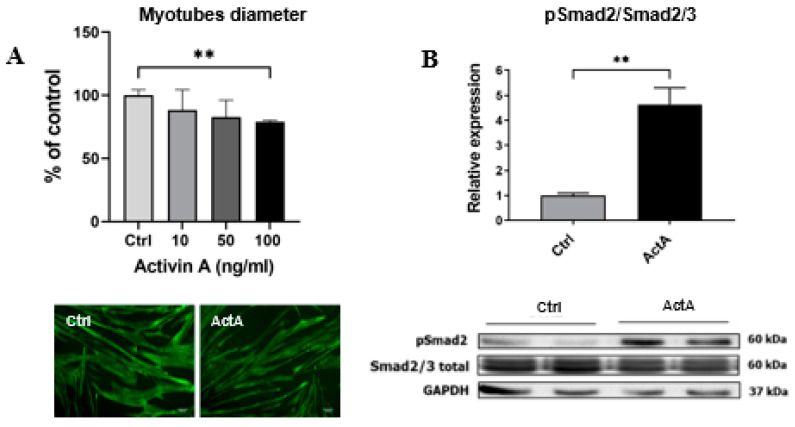
Activin A causes atrophy of human skeletal muscle cells. (**A**) Diameter of myotubes exposed to an increasing dose of recombinant activin A (ActA) or vehicle (Ctrl) for 48 h. Data are expressed as a percentage of the values for the control. (**B**) pSmad2/Smad2/3 ratio measured by western blot, in cells collected 1 h after exposure to recombinant ActA (100 ng/mL) or vehicle. Data are reported as means ± SEM (n = 3/condition). ** *p* < 0.01 vs. Ctrl.

**Figure 2 cells-11-01119-f002:**
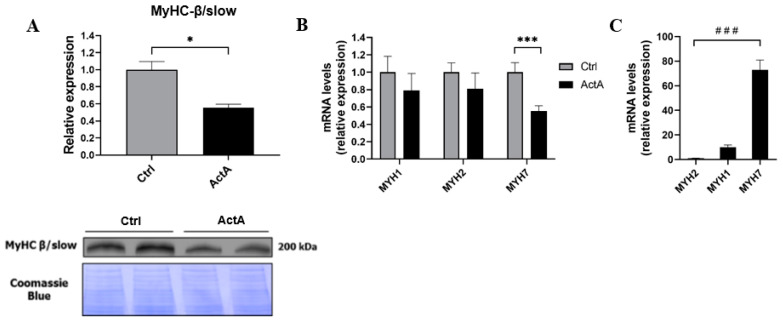
Activin A-induced myotube atrophy is characterized by a decrease in myosin-heavy chain-β/slow content. (**A**) Myosin-heavy chain-β/slow content measured by western blot, 48 h after exposure to recombinant ActA or vehicle. (**B**) *MYH1*, *MYH2*, and *MYH7* mRNA levels measured by RT-qPCR, 48 h after exposition to recombinant ActA or vehicle. (**C**) *MYH1*, *MYH2*, and *MYH7* relative expression in differentiated myotubes. Data are reported as means ± SEM (n = 3/condition). * *p* < 0.05 and *** *p* < 0.001 vs. Ctrl, ^###^ *p* < 0.001.

**Figure 3 cells-11-01119-f003:**
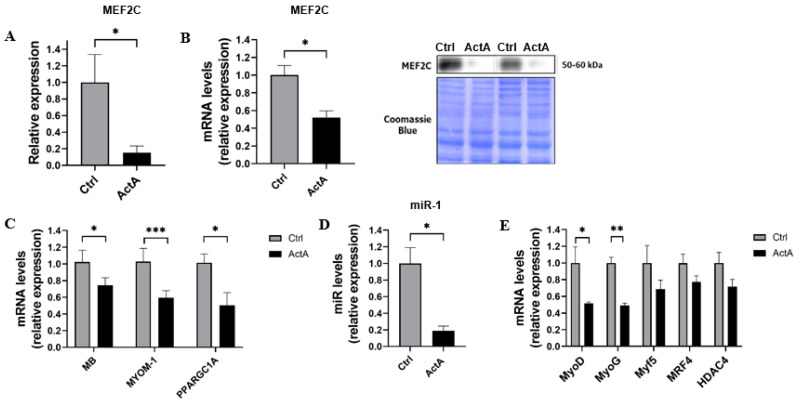
Activin A-induced myotube atrophy is associated with a downregulation of MEF2C expression and activity. (**A**) MEF2C content measured by western blot, (**B**,**C**,**E**) *MEF2C*, *MB*, *MYOM-1*, *PPARGC1A*, *MyoD*, *MyoG*, *Myf5*, *MRF4*, and *HDAC4* mRNA levels measured by RT-qPCR and (**D**) miR-1 expression, all measured 48 h after exposure to recombinant ActA or vehicle. Data are reported as means ± SEM (n = 5/condition). * *p* < 0.05, ** *p* < 0.01, and *** *p* < 0.001 vs. Ctrl.

**Figure 4 cells-11-01119-f004:**
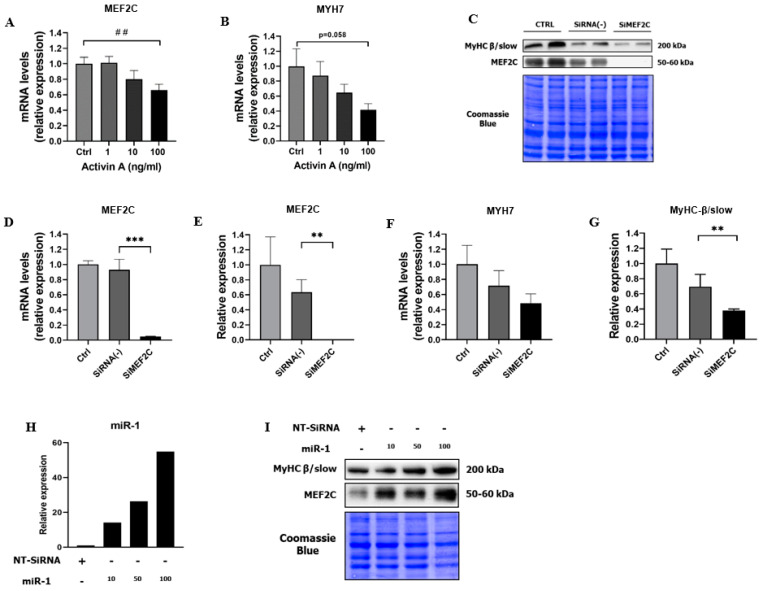
MEF2C is required to maintain MyHC-β/slow gene expression and protein content in differentiated myotubes. (**A**,**B**) *MEF2C* and *MYH7* mRNA levels measured by RT-qPCR 48 h after exposure to increasing the dose of recombinant ActA or vehicle. (**C**,**E**,**G**) MEF2C and myosin-heavy chain-β/slow content measured by western blot and (**D**–**F**) *MEF2C* and *MYH7* mRNA levels measured by RT-qPCR, 72 h after transfection of myotubes with a siRNA against MEF2C (SiMEF2C) or a negative control (non-targeting siRNA (SiRNA(−)). (**H**) MiR-1 expression measured by RT-qPCR and (**I**) MEF2C and myosin-heavy chain-β/slow content measured by western blot in myotubes, 48 h after transfection with miR-1 mimic (miR-1) in increasing concentrations (10,50,100 nM) or negative control (non-targeting SiRNA (NT-SiRNA)). Data are reported as means ± SEM (n = 3/condition), for miR-1 experiment (n = 1/condition). ^##^ *p* < 0.01, ** *p* < 0.01, and *** *p* < 0.001 vs. SiRNA(−).

**Figure 5 cells-11-01119-f005:**
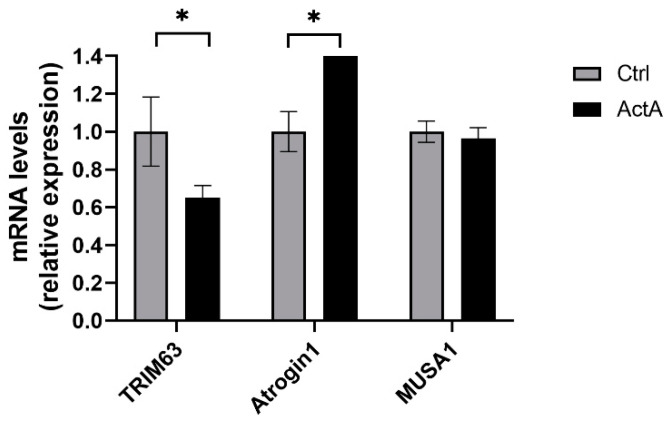
The activin A-induced myotube atrophy is not associated with an increase in classical E3 ubiquitin ligases. *TRIM63*, *Atrogin1*, and *MUSA1* mRNA levels measured by RT-qPCR, 48 h after exposure to recombinant ActA or vehicle. Data are reported as means ± SEM (n = 3/condition). * *p* < 0.05 vs. Ctrl.

**Figure 6 cells-11-01119-f006:**
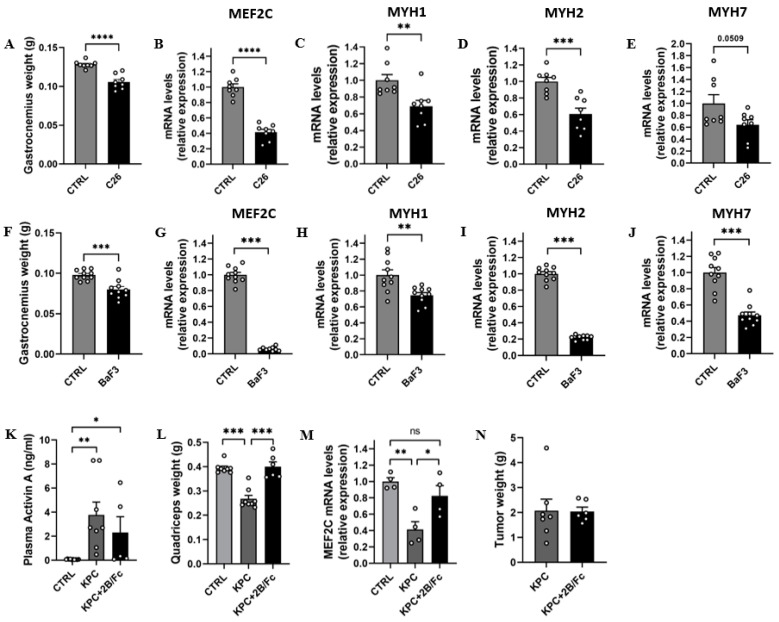
Cancer cachexia is associated with downregulation of muscle MEF2C expression and activity which is blunted by inhibition of activin A. (**A**) Gastrocnemius weight and (**B**–**E**) muscle *MEF2C, MYH1, MYH2*, and *MYH7* mRNA levels, 11 days after subcutaneous implantation of C26 cells (C26) or saline solution (Ctrl). (**F**) Gastrocnemius weight and (**G**–**J**) muscle *MEF2C, MYH1, MYH2*, and *MYH7* mRNA levels, 14 days after tail intravenous injection of BaF3 cells (Baf3) or saline solution (Ctrl). (**K**) Plasma ActA levels, (**L**) quadriceps weight, (**M**) muscle *MEF2C* mRNA levels, and (**N**) tumor weight, in KPC mice treated by ACVR2B/Fc (KPC + 2B/Fc) or phosphate buffered saline (KPC). Data are reported as means ± SEM (n = 4–10/groups). * *p* < 0.05, ** *p* < 0.01, *** *p* < 0.001, and **** *p* < 0.0001, ns: not significant.

**Figure 7 cells-11-01119-f007:**
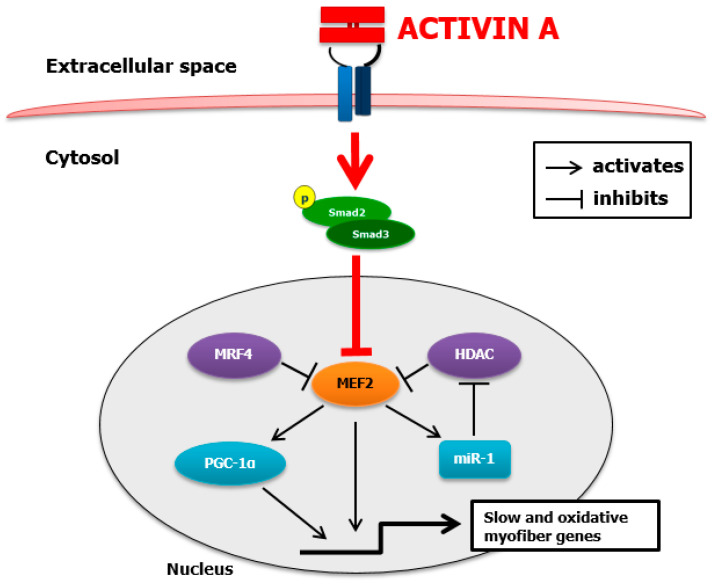
In muscle cells, activin A inhibits MEF2C expression and activity, leading to an inhibition of myosin-heavy chain-β/slow synthesis and muscle atrophy. Activin A, through activation and phosphorylation of Smad2/3, downregulates MEF2C expression and activity, leading to a decrease in MEF2C target expression (*MYH7*, peroxisome proliferator-activated receptor-γ activator-α (PGC-1α); microRNA (miR)-1), causing a decrease in MyHC-β/slow synthesis and muscle atrophy. ActA does not alter expression of negative regulators of MEF2C, such as myogenic regulatory factor 4 (MRF4) or class II histone deacetylases (HDAC).

**Table 1 cells-11-01119-t001:** Primer sequence used for amplification during real-time qPCR.

Gene	Primers 5′-3′		Accession No.
	Forward	Reverse	
**Human**			
*MYH7*	GAGCAAGCCAACACCAACCT	TGTGGCAAAGCTACTCCTCCATT	*NM_000257.3*
*MYH1*	TCCACTTTAAGGTCGCATCTCT	GTTCTGGGCTTCAATTCGCTC	*NM_005963.3*
*MYH2*	AGCCCTTGGAATGAGGCTGA	GCTCCGCCACAAAGACAGAT	*NM_017534.5*
*MEF2C*	TTCCAGTATGCCAGCACCG	GGCCCTTCTTTCTCAACGTCTC	*NM_002397.4*
*MB*	AGATTAAGCCCCTGGCACAGT	GATGCATTCCGAGATGAACTC	*NM_005368.2*
*MYOM-1*	GCAGCCTCAGCCTACGATTA	TGACATGCTTTTGACGTCCTG	*NM_003803.3*
*PPARGC1A*	CGGGATGATGGAGACAGCTA	CTTGGTGGAAGCAGGGTCAA	*NM_001354825.1*
*MyoD*	CGACGGCATGATGGACTACA	GGCAGTCTAGGCTCGACAC	*NM_002478.4*
*MyoG*	GCCATCCAGTACATCGAGCG	ATCTGTAGGGTCAGCCGTGA	*NM_002479.5*
*Myf5*	AACTACTATAGCCTGCCGGG	GATCCTGGAGAGGCAACCCA	*NM_005593.2*
*MRF4*	CTTGAGGGTGCGGATTTCCT	AAGCGCAGGCTCAGTTACTT	*NM_002469.2*
*HDAC4*	TTGGATGTCACAGACTCCGC	CCTTCTCGTGCCACAAGTCT	*NM_006037.3*
*TRIM63*	CATGTGCAAGGAGCACGAAG	GCCACCAGCATGGAGATACA	*NM_032588.3*
*Atrogin1*	TCACAGCTCACATCCCTGAG	AGACTTGCCGACTCTTTGGA	*NM_058229.3*
*MUSA1*	GTCATACTGCAGTGGGGGAAA	CGTGTCACACACATACATGGC	*NM_032145.4*
*GAPDH*	CGCTGAGTACGTCGTGGAGTC	GCAGGAGGCATTGCTGATGA	
**Mice**			
*MYH7*	GGTGCCAAGGGCCTGAATGAGGAG	GGTCTGAGGGCTTCACGGGCAC	
*MEF2C*	GCTGTTCCAGTACGCCAGCAC		*NM_025282.3*
*GAPDH*	TGCACCACCAACTGCTTA	GGATGCAGGGATGATGTTC	*NM_001289726.1*
*Tbp*			*Mm01277042_m1*

## Data Availability

The data presented in this study are available on request from the corresponding authors.

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
