# Peer review of "Activin A Causes Muscle Atrophy through MEF2C-Dependent Impaired Myogenesis"

_cells, 2022, doi:10.3390/cells11071119_

Round 1
Reviewer 1 Report
This manuscript written by Dr. Loumaye and colleagues aimed to clarify the effect of circulating Activin A (ActA) observed in cancer-induced cachexia (CC) on skeletal muscle, and found that down-regulation of MEF2C mediated by ActA could affect on muscle atrophy and MYH7 expression. This study is very interesting, and well examined.
However, there are several concerns. One of the critical concerns was that it was unclear whether MEF2C was a key molecule in ActA-induced muscle atrophy. This study showed that ActA down-regulated MEF2C and Myh7 expression in vitro, but provide no evidences that MEF2C directly affected muscle atrophy or Myh7 expression. The reviewer wonders whether MEF2C induced muscle atrophy or whether MEF2C-overexpression could prevent the muscle atrophy or the decline of MYH7 expression caused by ActA treatment.
Another concern was the effect of ActA and cachexia on myofiber type. The authors demonstrated that ActA treatment only affected gene expression in slow-twitch muscle, but not fast-twitch muscle. Additionally, the authors analyzed the gene expression of only MYH7 in CC-model mice. However, according to previous studies, inhibition of activin signaling using antibodies affected on both fast-type and slow-type muscle fiber (Lach-Trifilieff et al., 2014), and CC induced by C26 cells mainly affected on fast-type muscle fiber, but not on slow-type fiber (Acharyya et al., Cancer cell, 2005). Therefore, the authors' data are inconsistent with previous results, and need to be clarified these points.
The reviewer raised some concerns that need to be addressed in order to strengthen the conclusions drawn by the authors.
In line 201, the authors described that the effect of ActA on myotube size was dose-dependent, but that data was not presented. Since this data is important for understanding the extent of ActA-induced muscle atrophy, please provide the results. Additionally, the authors mentioned that the chosen dose of ActA (100ng/ml) had no effect on cell viability. However, the reviewer wonders how the group evaluate cell viability during differentiation. Please provide the method how the cell viability was assessed.
In figure 2B, the authors analyzed the effect of ActA on muscle fiber type, and found that ActA affected only the expression of slow muscle-related genes, but not the fast muscle. However, previous studies showed that inhibition of activin type 2 receptors induced muscle hypertrophy in both fast and slow muscle (Lach-Trifilieff et al., 2014). The reviewer wonders how the authors interpret this difference.
In figure 3E, the expressions of myogenic factors including MyoD and Myogenin were suppressed by ActA-treatment. How was the expression of these genes in animal models of CC?
In figure 4, the authors have demonstrated that MEF2C and MYH7 expressions were dose-dependently declined by ActA-treatment, and that suppression of MEF2C by siRNA decreased MYH7 expression. The authors should confirm whether MEF2C-knockdown induce muscle atrophy by immunohistochemistry.
Additionally, these results were insufficient to show the evidence that MEF2C directly affected on MYH7 expression. In order to confirm whether MEF2C is key factor for muscle atrophy and MYH7 expression, the group should confirm whether overexpression of MEF2C could prevent muscle atrophy and the reduction of MYH7 expression induced by ActA-treatment.
In figure 6A and D, the authors measured the weight of the gastrocnemius muscles, which were enriched in fast-twitch muscle fibers, in a CC model mouse. If the authors believe that ActA acted specifically on slow-type muscles, the weight of the slow-type soleus muscles also should be measured.
In figure 6C and F, the authors evaluated the expression of slow-type MYH7 expressions, and showed the decrease of MYH7 expression in gastrocnemius muscle from CC-model mice. The reviewer requests to show the expressions of fast-type MYH1 and MYH2 gene, as well. According to previous study, atrophic fibers due to CC were selectively fast-type in origin while slow-type fibers remained relatively unchanged (Acharyya et al., 2005). These are inconsistent with the authors results, and please discuss about this discrepancy.
Author Response
Dear reviewers,
Please see the attachment to find our answers to the comments.
Best regards
Audrey Loumaye

Reviewer 2 Report
This work analyze the role of Activin A in muscle atrophy and its involvement in the expression the MyHC b/slow. The authors demonstrated that Activin A inhibits myogenesis and MyHC-b/slow synthesis by down regulating MEF2C, the main transcription factor of MyHC-b/slow. Activin A also targets miR-1 expression, a positive regulator of MEF2C expression.
The results are convincing and clearly presented. It's a serious and innovating work.
Minor points:
Lines 20, 366, 513: Why do the authors explain that the slow myosin is the main isoform of human muscles? This isoform is only specific of slow-twitch muscles?
Author Response
Dear reviewers,
Please see the attachment to find our answers to comments
Best regards
Audrey Loumaye

Round 2
Reviewer 1 Report
The authors have adequately addressed all comments.